# Role of the CB2 Cannabinoid Receptor in the Regulation of Food Intake: A Systematic Review

**DOI:** 10.3390/ijms242417516

**Published:** 2023-12-15

**Authors:** Luis Miguel Rodríguez-Serrano, María Elena Chávez-Hernández

**Affiliations:** Facultad de Psicología, Universidad Anáhuac México, Universidad Anáhuac Avenue #46, Lomas Anáhuac, Huixquilucan 52786, Mexico; mariele_chavez@yahoo.com

**Keywords:** cannabinoid 2 receptor, food intake, overweight, systematic review

## Abstract

The CB2 cannabinoid receptor has been found in brain areas that are part of the reward system and has been shown to play a role in food intake regulation. Herein, we conducted a systematic review of studies assessing the role of the CB2 receptor in food intake regulation. Records from the PubMed, Scopus, and EBSCO databases were screened, resulting in 13 studies that were used in the present systematic review, following the PRISMA guidelines. A risk of bias assessment was carried out using the tool of the Systematic Review Center for Laboratory Animal Experimentation (SYRCLE). The studies analyzed used two main strategies: (1) the intraperitoneal or intracerebroventricular administration of a CB2 agonist/antagonist; and (2) depletion of CB2 receptors via knockout in mice. Both strategies are useful in identifying the role of the CB2 receptor in food intake in standard and palatable diets. The conclusions derived from animal models showed that CB2 receptors are necessary for modulating food intake and mediating energy balance.

## 1. Introduction

Obesity is a public health issue since several related chronic diseases, such as hypertension, diabetes, and cardiovascular disease, have adverse effects [1]. In this regard, Silventoinen and Konttinen [2] suggested that obesity is a neuronal and behavioral disease with a solid genetic background mediated primarily by eating behaviors and environmental sensitivity. Furthermore, the classical view is that homeostatic feeding is necessary for basic metabolic processes and survival. In contrast, sensory perception or pleasure drives hedonic feeding (intake of palatable food) [3]. In this regard, food properties have a central role in intake; in particular, the palatability of food can lead to the development of obesity in susceptible individuals [4], produce metabolic syndrome and cognitive impairment [5], and enhance food intake by hedonic mechanisms that prevail over caloric necessities [6]. Therefore, the overconsumption of palatable food is a risk factor that can lead to non-homeostatic feeding, and it can also outweigh those of caloric need and contribute significantly to obesity [7,8]. Additionally, individuals with obesity respond to palatable food without attention to the consequences of a hypercaloric diet, which reflects behavioral impulsiveness [9]. It was also shown that the consumption of palatable foods can have an anxiolytic effect [10]. Furthermore, it has been suggested that hypercaloric intake can alter synaptic plasticity in the mesolimbic system in animal models [11] and humans [12].

The consumption of highly palatable foods involves learned habits and preferences through the reinforcing properties of powerful and repetitive rewards, which may lead to dysfunction of the neurobiological mechanisms that regulate eating behaviors [13,14,15]. Furthermore, the roles of crucial structures, such as the anterior cingulate, amygdala, insular cortex, nucleus accumbens (NAC), hippocampus, and hypothalamus, in food intake have been identified [16]. The endogenous cannabinoid system (ECS) acts in the mesolimbic system pathway and is involved in food reward [6]. It was shown that the consumption of palatable foods reduced the expression of endocannabinoid production enzymes and their receptors [17]. In addition, the ECS can modulate dopaminergic synaptic transmission in palatable food intake [18] and is a potent regulator of feeding [19]. Studies have shown that it is expressed in brain areas that are involved in the modulatory brain loops in food intake and in reward areas in body weight disorders in humans [20]. There is a growing interest in exploring the mechanisms underlying the palatability of food and how they can induce obesity. In this regard, obesity and eating disorders have been associated with endocannabinoid signaling, and it has been suggested that the ECS has a role in reorienting energy balance towards energy storage [21]. Also, it has been shown that there is interaction between the ECS and other systems, such as the serotonergic and GABAergic systems, in regulating eating behavior [22]. Additionally, a study by Bello et al. [23] showed a significant decrease in CB1 receptor expression in the cingulate cortex in rats with access to palatable food.

### 1.1. The Endocannabinoid System

The ECS is comprised of two cannabinoid receptors, the endogenous ligands or endocannabinoids: anandamide (AEA) and 2-arachydonyl glycerol (2AG), and the enzymatic machinery in charge of synthesis and degradation [24,25]. The first endocannabinoid was identified in 1992 from pig brains and named arachidonylethanolamide, or anandamide (AEA) [26]. A few years later, in 1995, 2-arachidonyl glycerol (2-AG), a second endocannabinoid, was discovered from canine intestines [24] and later its presence was described in the brain [27]. Unlike other neurotransmitters, endocannabinoids are also synthesized on demand, depending on increased intracellular Ca^2+^ [28].

AEA is synthesized from phosphatidylethanolamine, which is transformed into N-arachidonoyl phosphatidylethanolamine by the action of N-acetyltransferase, and finally into N-arachidonoyl ethanolamine (anandamide, AEA) by phospholipase-D. AEA is finally degraded by the fatty-amino-acid-hydrolase (FAAH) into arachidonic acid and ethanolamide [29]. 2AG is synthesized from phosphatidylinositol that is transformed into diacylglycerol by the actions of phospholipase-C and finally into 2-AG by DAG-lipase. 2AG is degraded by the enzyme monoacylglycerol-lipase (MAGL) into arachidonic acid and glycerol [30].

The ECS also includes two cannabinoid receptors: CB1 and CB2. The CB1 receptor was first described in 1988 by Devane et al. [26] in the central nervous system (CNS). In 1993, the CB2 receptor was first described in peripheral tissues [31] and, several years later, its presence was detected in the CNS [32]. In this regard, the CB1 receptor has shown expression in the CNS and peripheral tissues [33], whereas CB2 is expressed mainly in peripheral tissue and immune cells and was later described in neurons [32,34,35,36] in areas such as the VTA [37], and neuroglia [38,39].

Both CB1 and CB2 receptors are G-Protein-coupled receptors (GPCR) that, when activated, inhibit adenylyl cyclase and activate mitogen-activated protein kinase (MAPK) by signaling through Gi/o proteins [40,41]. CB1 receptors modulate ion channels, inhibiting voltage-dependent Ca^2+^ channels and activating inwardly rectifying K^+^ channels [42]. Additionally, CB2 receptors are mainly expressed in postsynaptic neurons [37,43], while CB1 receptors are predominantly expressed in neuronal presynaptic terminals, suggesting opposite roles of CB1 and CB2 receptors in the regulation of neuronal firing and neurotransmitter release [44]. Furthermore, the ECS is a retrograde regulator of synaptic neurotransmission, which modulates the activity of different neurotransmitters such as GABA, glutamate, and dopamine [33,37,45,46]. Also, in the mesolimbic pathway, some forms of long-term plasticity are facilitated by ECS retrograde signaling.

### 1.2. Cellular Mechanisms of the CB2 Receptor

Evidence from electrophysiology studies indicates that CB2 receptor activation alters neuronal activity and excitability [45,47]. Given that the CB2 receptor couples to Gi/o proteins, their activation is associated with several different cellular pathways, such as adenylate cyclase (AC), cAMP, protein kinase A (PKA), ERK 1/2, p38 MAPK p38 (p38 MAPK), and AKT [40]. Notably, the CB2 receptor has more affinity to Gi than to Go proteins [40,45]. CB2 receptor activation results in a mediated inhibition of AC activity and subsequent closure of Ca^2+^ and opening of K^+^ channels and stimulates MAPK cascades [45,48,49], specifically, ERK and p38 [40,45,50], and overall resulting in the inhibition of neuronal activity.

Furthermore, evidence shows that CB2 receptors are expressed in dopamine (DA) neurons [51], with recent research showing that CB2 receptors are expressed in the midbrain DA neurons in mice, and that activation of CB2 receptors in the ventral tegmental area (VTA) inhibits DA neuronal firing [52,53]. For example, studies show that acute exposure to the CB2 receptor agonist (JWH-133) reduces DA neuronal excitability in the VTA [54], while this effect can be blocked by AM630, a selective CB2 receptor antagonist [53]. In this regard, it has been shown that CB2 receptors regulate DA 2 receptor expression [55,56]. Additionally, JWH-133 has been reported to produce inhibition in VTA dopamine neuronal activity and NAC dopamine release [37,53].

Therefore, CB2 receptors can modulate synaptic plasticity in reward pathways [53,57]. Additionally, it has been shown that chronic administration of the CB2 receptor antagonist (AM630) induces anxiolytic-like effects and modifies GABA activity in the cingulated cortex and amygdala [55]. Verty et al. [58] show that chronic systemic administration of a CB2 receptor agonist (JWH-015) increases PKA expression, which could stimulate PKA activity to inhibit lipogenesis and increase lipolysis, resulting in reduced body weight. These results show that CB2 receptors modulate the release of DA in mesolimbic neurons and regulate hedonic behaviors such as the consumption of palatable food.

Considering the importance of the ECS in several brain regions involved in food intake, the role of the CB2 receptor in food intake needs to be defined; in this regard, we conducted a systematic review to identify, evaluate, and summarize the findings from animal studies assessing the role of the CB2 receptor in food intake regulation.

## 2. Methods

### 2.1. Information Sources and Search Strategy

Following the PICO (population, intervention, comparison, outcome) research question format, elements stated for the present systematic review are as follows: Population: rats or mice (animal models), intervention: administration of synthetic CB2 selective agonists/antagonist or knockout of the CB2 receptor, comparison: control/vehicle group or wild-type group, outcome measure: effect on food intake. Furthermore, a systematic search was conducted following the guidelines for systematic reviews and meta-analyses (PRISMA) [59,60]. Three databases were searched: PubMed, EBSCO, and Scopus. The following keywords were used for the search based on Medical Subject Headings (MeSH) terms: “Eating”, “Food Intake”, “Feed Intake”, “Obesity”, “Overweight”, “CB2 Receptor”, “Murinae”. These descriptors were searched in the title and abstract domains.

Scientific articles in English were included with no restrictions regarding publication dates. The inclusion criteria were peer-reviewed papers and use of rodent models of food intake. Articles were excluded based on the following criteria: review articles, systematic review and/or meta-analysis articles, and full text not available.

### 2.2. Data Extraction and Assessment of Risk of Bias

All studies were extracted by one author (R.S.) and checked by a second author (C.H.). The identification and extraction of the papers in this systematic review was carried out using PRISMA. In addition, the risk of bias was assessed by two reviewers (R.S. and C.H.) using the Systematic Review Center for Laboratory Animal Experimentation (SYRCLE) tool [61].

## 3. Results

### 3.1. Search Results

A total of 186 studies were identified from three databases; 140 duplicates were removed, and 46 records were included for title and abstract screening. Of those, 16 were included due to full-text availability. After this full-text screening, 13 studies were included in the systematic review (Figure 1).

### 3.2. Risk of Bias Assessment

The risk of bias assessment results are presented in Table 1. The risks of selection bias, performance, detection, attrition, information, and others were evaluated and divided into the following ten questions included in the SYRCLE tool [61]:

Q1: Was the allocation sequence adequately generated and applied?

Q2: Were the groups similar at baseline or were they adjusted for confounders in the analysis?

Q3: Was the allocation adequately concealed?

Q4: Were the animals randomly housed during the experiment?

Q5: Were the caregivers and/or investigators blinded from knowledge which intervention each animal received during the experiment?

Q6: Were animals selected at random for outcome assessment?

Q7: Was the outcome assessor blinded?

Q8: Were incomplete outcome data adequately addressed?

Q9: Are reports of the study free of selective outcome reporting?

Q10: Was the study apparently free of other problems that could result in high risk of bias?

**Table 1 ijms-24-17516-t001:** Risk of bias assessment using SYRCLE’s risk of bias tool for animal studies.

Study	Q1	Q2	Q3	Q4	Q5	Q6	Q7	Q8	Q9	Q10
Werner 2003 [62]	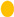	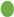	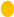	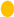	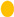	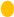	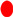	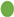	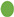	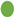
Onaivi 2008 [63]	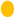	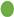	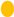	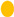	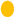	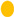	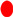	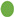	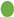	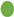
Agudo 2010 [64]	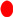	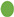	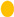	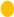	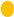	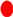	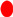	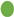	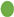	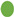
Ishiguro 2010 [65]	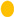	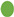	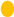	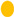	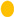	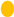	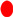	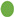	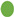	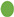
Ting 2015 [66]	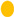	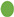	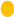	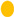	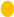	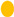	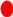	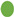	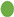	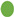
Verty 2015 [58]	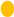	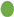	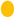	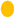	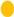	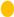	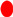	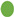	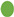	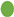
Schmitz 2016 [67]	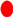	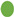	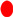	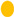	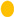	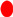	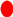	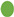	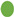	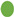
Diaz-Rocha 2018 [68]	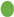	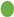	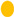	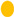	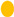	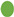	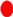	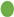	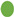	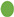
Alshaarawy 2019 [69]	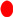	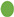	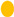	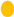	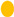	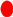	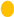	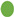	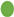	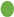
Bi 2019 [70]	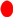	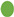	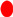	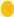	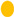	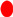	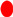	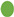	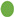	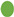
Bourdy 2021 [71]	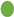	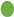	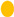	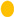	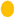	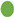	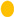	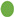	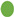	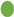
De Ceglia 2023 [17]	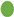	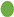	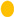	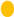	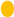	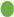	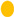	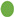	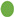	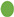
Rorato 2023 [72]	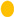	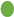	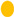	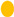	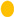	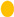	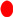	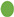	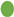	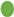

Note: low risk of bias (
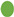
); high risk of bias (
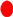
); unclear risk of bias (
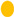
).

### 3.3. The Role of the CB2 Cannabinoid Receptor in Food Intake

The studies included in the present systematic review are summarized in Table 2 and were published between 2003 and 2023. Eight studies used mice (61.5%), three used Wistar rats (23.1%), one used Sprague Dawley rats (7.6%), and one used Lewis rats (7.6%) as subjects. In addition, 76.9% of studies used male subjects, while 23.1% used male and female subjects. All of the studies used adult mice or rats. Eight of the thirteen included studies (61.5%) evaluated the administration of agonists or antagonists of the CB2 receptor via an intraperitoneal or intracerebroventricular route, while five studies (38.5%) evaluated the expression of the CB2 receptor. Seven studies (53.8%) evaluated the CB2 receptor’s role in the intake of a standard diet (chow), while six studies used a palatable diet (46.2%).

#### 3.3.1. Administration of CB2 Agonists and Antagonists

Several CB2 receptor ligands have been developed to therapeutically target this receptor. In this regard, the four ligands used in the studies included in the present review are shown Table 3, indicating their structure, action mechanism, CB2 receptor affinity (pKi), and the selectivity of agonists and inverse-agonists used in the studies based on mouse CB2 receptor studies.

The oral sucrose self-administration test is used to study feeding and binge eating in animal models. Using this test, Bi et al. [70] showed that pharmacological blockade or genetic deletion of the CB2 receptor blocked the pharmacological action of cannabidiol in sucrose self-administration. They also showed that the administration of JWH-133 (a selective CB2 receptor agonist) produced a reduction in sucrose self-administration, suggesting that the CB2 receptor has a role in controlling food intake. In addition, the chronic systemic administration of JWH-015 (a CB2 receptor agonist) significantly reduces food intake, body weight gain, and adipocyte cell size in diet-induced obese mice [58]. However, Rorarato et al. [72] found that the intracerebroventricular administration of HU308 (a CB2 receptor agonist) did not modify food intake.

On the other hand, it has been shown that the administration of a CB2 receptor antagonist (AM630) induces an increase in food intake after food deprivation [62,65,66] and when food is available ad libitum [63]. In addition, Agudo et al. [64] showed that the administration of an antagonist (SR 144528) to wild-type mice does not modify chow intake.

#### 3.3.2. Depletion and Expression of the CB2 Receptor

Alshaarawy et al. [69] evaluated the effect of a high-fat diet on weight gain in CB2 receptor knockout, CB1and CB2 double-knockout, and wild-type mice. Their results showed that the weight gain was similar in both the CB2 receptor knockout and wild-type mice groups, while in the double knockout mice, weight gain was significantly lower; these findings indicate that lacking both CB1 and CB2 receptors protects mice from diet-induced obesity, while the absence of the CB2 receptor only does not protect in this aspect when compared to wild-type mice. Additionally, Schmitz et al. [67] showed that a CB2 receptor deficiency reduces feeding behavior, but also that it induces obesity in adult 15- to 18-month-old mice. Moreover, Dias-Rocha et al. [68] evaluated the effects of a maternal high-fat diet on male and female rat offspring, at weaning and adulthood, on CB2 receptor expression in the hypothalamus and high-fat food intake, and found an increase in the expression of hypothalamic CB2 in female offspring, and also a higher preference for high-fat diets in male and female offspring.

Furthermore, Bourdy et al. [71] showed that there was an increase in CB2 receptor expression in the NAC after exposure to a freely available high-sugar diet. In addition, de Ceglia et al. [17] showed increased CB2 receptor expression in the prefrontal cortex of adult rats after chronic exposure to a cafeteria diet. Nevertheless, intracerebroventricular administration of an agonist (HU308) did not modify the intake of a high-fat diet in adult mice [72]. In summary, these studies identified a role for the CB2 receptor in food intake through depletion of the receptor and exposure to several diets, which resulted in increased CB2 receptor expression and food intake.

## 4. Discussion

The present systematic review shows compelling evidence from animal studies of CB2 receptors’ role in regulating food intake. The PICO research question stated for the present systemic review considered the following elements: Population: rats or mice (animal models), intervention: administration of synthetic CB2 selective agonist/antagonist (intraperitoneal or intracerebroventricular), or knockout of CB2 receptor, comparison: control/vehicle group or wild-type group, outcome measure: effect on food intake. The results show consistent evidence from animal studies that the CB2 receptor could play a role in the rewarding effects of palatable food through distinct neuronal mechanisms and that the intake of palatable food induces increased expression of CB2 in the NAC and VTA [71]. Additionally, recent findings reveal the expression of CB2 receptors in brain areas that are integral to the reward system [37,53,76].

The consumption of highly palatable foods involves learned habits and preferences that reinforce the properties of powerful and repetitive rewards, which leads to the dysfunction of the neurobiological mechanisms that regulate eating behaviors [13,14,15]. For instance, the hedonic response to food might depend on the ECS through modulation of the mesocorticolimbic dopamine system [77]. The results from animal models show that the ECS modulates neuronal plasticity changes that result from excessive palatable food consumption.

The CB2 receptor is present in reward system brain areas [37,53,76] such as the mesocorticolimbic system and its activity has been shown to modulate the release of dopamine, therefore regulating hedonic behaviors [77]. In addition, recent research indicates that activation of CB2 receptors expressed in VTA dopamine neurons in mice inhibits dopamine-induced neuronal firing [52,53]. These results indicate that CB2 receptors have a role in the rewarding effects of palatable food through distinct mechanisms [71,72]. Consistent with these findings, it has been shown that ablation of the CB2 receptor led to increased food intake, body weight gain, and adipose tissue hypertrophy in mice [64,67,78]. Nevertheless, Romero-Zerbo et al. [79] showed that specific overexpression of CB2 receptors in the brain induced hyperglycemia and decreased food intake. Additionally, Lillo et al. [80] showed a functional and molecular interaction between ghrelin and CB2 receptors in striatal neurons in offspring of mothers fed a high-fat diet.

Several studies have indicated that CB2 receptors are a crucial target in diseases such as drug-use disorders and that their functional manipulation may regulate ethanol motivation and consumption vulnerability. For instance, one study suggested that reducing the expression of the gene encoding for CB2 receptors (CNR2) in the mesocorticolimbic system increased mice’s risk of developing excessive ethanol consumption [81]. The present review showed the recent evidence for the CB2 receptor’s role in regulating the rewarding effects of palatable food intake.

Studies have shown that CB2 receptor signaling inhibits feeding behavior and that CB2 receptor ligands can be a therapeutic target for eating disorders and obesity [82]. Studies with animal models showed that selective CB2 receptor agonists modulate food intake, reduce weight gain, relieve glucose intolerance, and enhance insulin sensitivity [44,58,67,83,84]. On the other hand, other studies have indicated that CB2 receptor ablation leads to increased food intake, body weight gain, and adipose tissue hypertrophy in mice [64,67,78]. Likewise, García-Blanco et al. [85] show that depletion of CB2 receptors increases vulnerability to develop palatable food addiction. Furthermore, it has been shown that the administration of JWH-133 reduces body weight gain, relieves glucose tolerance, and enhances insulin sensitivity in mice models of obesity [44]. Additionally, it has also been shown that chronic systemic administration of JWH-015 (a CB2 agonist) significantly reduces food intake, body weight gain, and adipocyte cell size in diet-induced obese mice [58]. Also, Bermudez-Silva et al. [83] showed that the administration of CB2 receptor agonists increases glucose tolerance, an effect that is opposite of those shown with CB2 receptor antagonists. Furthermore, it has been suggested that CB2 receptor agonists (JW-133) have a role in anti-inflammatory and anti-obesity mechanisms [44,82]. Additionally, Alshaarawy et al. [69] showed that weight gain was similar in both CB2 knockout and wild-type mice groups, while double knockout mice weight gain was significantly lower, indicating that the absence of the CB2 receptor only does not protect in this aspect when compared to wild-type mice.

Dias-Rocha et al. [68] evaluated the effects of a maternal high-fat diet on male and female rat offspring at weaning and adulthood, and reported an increase in CB2 receptor expression in the hypothalamus and high-fat food intake. Given that CB2 is widely expressed in immune cells and glia, this receptor may play a role in initiating inflammation at the central level. In this regard, the authors suggest that the heightened expression of CB2 in the hypothalamus of female offspring from dams exposed to a high-fat diet may contribute to the triggering of central inflammation. Since hypothalamic inflammation is closely associated with the consumption of a high-fat diet, the authors conclude that the elevated levels of CB2 in female offspring might be a contributing factor in predisposing them to adult hyperphagia and overweight.

Regarding the CB2 receptor’s influence in the modulation of obesity and of adiposity activity and morphology, in a study by Rossi et al. [86] genetic analysis was conducted on genomic DNA extracted from obese children and adolescents. The results showed that the CB2-Q63R variant in obese children is associated with a high z-score body mass index. Additionally, in vitro analysis of subcutaneous adipose tissue biopsies from lean and obese subjects showed that CB2 blockade with the AM630 reverse agonist increased inflammatory adipokine release and fat storage and reduced browning. On the other hand, CB2 agonist JWH-133 reversed all of the obesity-related effects. These results indicate that the CB2 receptor may represent a pharmacological target for reducing obesity-related inflammatory states and excessive fat storage.

This evidence indicates that CB2 receptor activity is critical to modulating food intake and may also be essential to restoring the neurobiological imbalance that results from palatable food overconsumption, providing a novel target for treating obesity and eating disorders, such as binge eating disorders. In addition, CB2 receptor ligands may provide a novel treatment for managing overweight and obesity, conditions which also involve reward and mesocorticolimbic system activity.

Some issues have emerged when studying CB2 receptor activity and its association with palatable food intake. The first is its low expression levels in the CNS and the lack of reliable antibodies [87]. Despite this, evidence from studies with electrophysiological and molecular biology techniques provides strong and direct evidence for the neuronal expression of functional CB2 receptors in the CNS [57]. Another limitation is that the number of studies focusing on selective CB2 receptor activity is still scarce relative to those focusing on CB1 receptor activity. We are just starting to understand the role that CB2 receptors play in regulating the consumption of palatable food. Nevertheless, their activity could provide a novel, relevant target for the clinical treatment of palatable food overconsumption and obesity. Additionally, some variables should be considered when interpreting results from the studies included, such as, antibody specificity, use of ligands, and upregulation of CB1 in CB2 KO mice (usually not tested). Nevertheless, results from the studies included in the present systematic review show promising evidence of the role the CB2 receptor plays in regulating food intake. Therefore, more studies controlling for these potentially confounding variables are necessary in order to fully understand the role of the CB2 receptor in food intake regulation.

## 5. Conclusions

In this review, we present a systematic analysis of the CB2 receptor’s role in food intake regulation, as identified through rodent studies. This systematic review identifies, evaluates, and summarizes preclinical evidence from animal studies that indicate the involvement of CB2 receptors in food intake regulation. Understanding this aspect is crucial in comprehending the underlying mechanisms and in proposing novel treatments for overweight and obesity.

Research has focused on two main approaches: (1) the administration of an agonist or antagonist via intraperitoneal and intracerebroventricular routes; and (2) depletion of CB2 receptors. Both strategies are useful approaches for investigating the role of the CB2 receptor in food intake, with either standard or palatable diets. Overall, the evidence shows that CB2 receptors in CNS are necessary for modulating food intake and mediating energy balance.

## Figures and Tables

**Figure 1 ijms-24-17516-f001:**
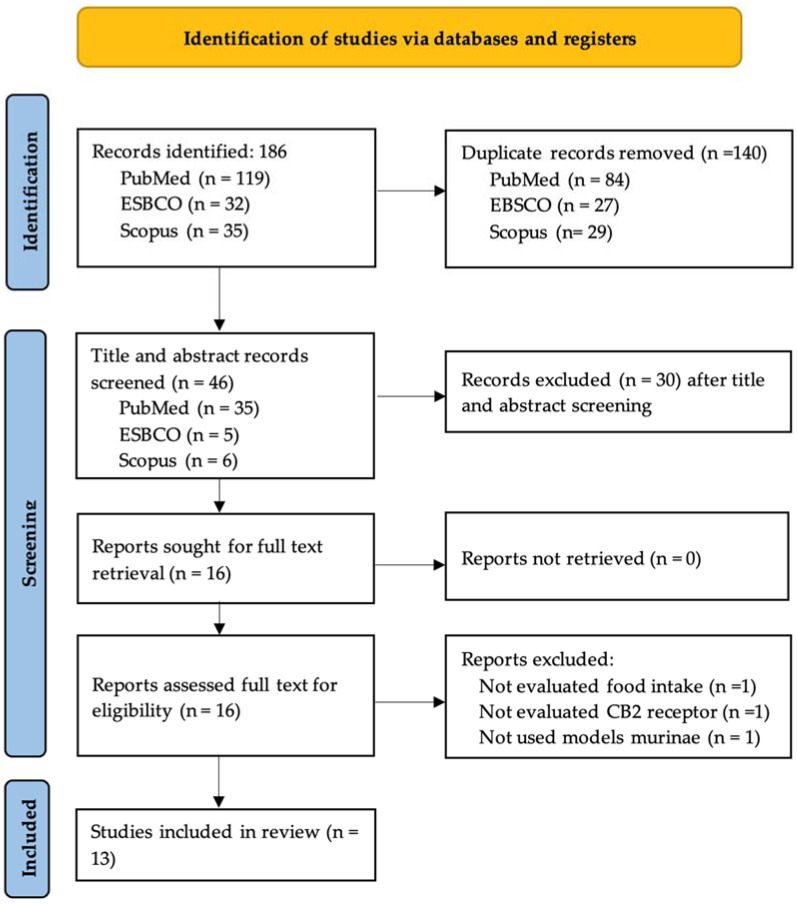
Flow diagram of study selection.

**Table 2 ijms-24-17516-t002:** Description of the studies investigating the CB2 cannabinoid receptor in food intake.

Study	Animal Model Characteristics	Treatment	Results
Werner 2003 [62]	Male Lewis rats (220–240 g)	Administration (i.c.v.) of antagonist (AM630) at different doses: 2.5, 5, 10, and 25 µg	↑ Food intake (chow)
Onaivi 2008 [63]	Male mice: C57Bl/6J, Balb/c, and DBA/2 strains	Acute administration (i.p.) of antagonist (AM630) at 10 mg/kg	The C57BL/6 mice and DBA/s mice ↓ food intake (chow) The Balb/c mice = food intake (chow)
Agudo 2010 [64]	Male knockout mice (Cb2^−/−^) and wild-type littermates (Cb2^+/+^); ages: 2, 6, and 12 months	Wild-type received chronic administration (i.p.) of antagonist (SR 144528) at 3 mg/kg	CB2R knockout mice ↑ food intake (chow) Wild-type mice = food intake (chow)
Ishiguro 2010 [65]	C57Bl/6J male and female mice	Acute administration (i.p.) of antagonist (AM630) at 10 mg/kg	↑ Food intake (chow)
Ting 2015 [66]	Male Sprague Dawley rats (240 to 310 g)	Acute administration (i.p.) of antagonist (AM630) at different doses: 0.3, 1, and 3 mg/kg	↑ Food intake (chow)
Verty 2015 [58]	Male C57BL/6 mice (8 weeks old)	Acute administration (i.p.) of agonist (JWH-015) at different doses: 1.0, 5.0, or 10.0 mg/kg. Co-administered of agonist (JWH-015, 10 mg/kg) with antagonist (AM630, 5 mg/kg)	↓ Food intake (chow) with administration of JWH-015 (10 mg/kg) = food intake (chow) when co-administered agonist and antagonist
Schmitz 2016 [67]	Male and female knockout mice (Cb2^−/−^) (1.2–1.8 years old)	CB2 receptor deficiency to study age-associated obesity	↓ Food intake (chow) CB2^−/−^ mice became obese.
Dias-Rocha 2018 [68]	Male and female Wistar rats (170 days old)	Maternal high-fat (HF) diet to study effects in rat offspring	↑ Food preference (high-fat diet) in males and females Females ↑ expression of CB2 receptor
Alshaarawy 2019 [69]	Male knockout mice (Cb2^−/−^) (8 weeks old)	12 weeks on low-fat or high-fat diet	↑ Weight gain on high-fat diet, but not different from wild-type mice = Food intake (high-fat diet)
Bi 2019 [70]	Male knockout mice (Cb2^−/−^) (8 to 14 weeks old)	Acute administration (i.p.) of agonist (JWH-015) at different doses: 10 and 20 mg/ kg	↓ Sucrose self-administration in wild-type mice, but not knockout mice, with doses of 10 and 20 mg/ kg
Bourdy 2021 [71]	Male Wistar rats (200 ± 10 g)	6 weeks of free-choice regimen of high-fat or sugar diet	↑ Expression of CB2 receptor in NAC due to high sucrose
De Ceglia 2023 [17]	Adult male Wistar rats (280–300 g)	40 days of exposure to palatable cafeteria diet	↑ Expression of CB2 receptor in prefrontal cortex
Rorato 2023 [72]	Male C57BL mice (7–8 weeks old)	56 days of exposure to high-fat diet. Chronic administration (i.c.v.) of agonist (HU308) at different doses: 1.0, 5.0, or 10.0 mg/kg	= Food intake (high-fat diet)

↑ indicates increase, ↓ indicates decrease, i.p. = intraperitoneal, i.c.v. = intracerebroventricular.

**Table 3 ijms-24-17516-t003:** Structure, affinity (pKi), and selectivity on mouse CB2R of selective ligands used in the included studies.

Ligand	Structure	Action Mechanism	pKi on CB2R	CB2R Selectivity *
HU 308	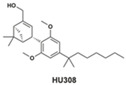	CB2-selective agonist	7.15 [73] 7.0 [74]	12 [73]
JWH-015	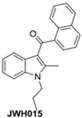	Moderately CB2-selective agonist	6.63 [73] 6.5 [74]	5 [73]
AM 630	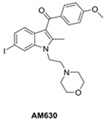	CB2-selective inverse agonist	7.66 [73] 7.7 [74]	115 [73]
SR144528	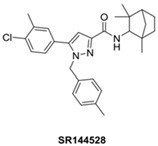	CB2-selective inverse agonist	10.7 [73,75] 10.5 [74]	6026 [73]

* CB2R selectivity calculated as follows: 10^(pKi CB2R − pKi CB1R).

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
