# Peer review of "Role of the CB2 Cannabinoid Receptor in the Regulation of Food Intake: A Systematic Review"

_ijms, 2023, doi:10.3390/ijms242417516_

Round 1

Reviewer 1 Report

Comments and Suggestions for Authors

The manuscript was well-written but there are several areas that need to be revised before consideration for publication. 

1. Missing the structures of select CB2 ligands and their Structure-activity-relationship studies. 

2. Any new modalities targeting CB2 receptors? Any drug discovery campaigns going on in the biotech/pharma industry?    

3. Any other indications of modulating CB2 receptor being pursued? and the potential risk/benefit of targeting CB2 receptor for obesity? 

Comments on the Quality of English Language

The English language is scientific and clear. 

Author Response

Dear Reviewer,

Thank you for your valuable feedback on the paper. Regarding your observations, the following changes and additions have been included in the manuscript

Missing the structures of select CB2 ligands and their Structure-activity-relationship studies. 

  • A table including structure, action mechanism and receptor affinity of ligands used in the included studies has been added to the results section.

Any new modalities targeting CB2 receptors? Any drug discovery campaigns going on in the biotech/pharma industry?

  • To our knowledge, no new modalities to target CB2, other than agonists and antagonists/inverse agonists, have been reported in scientific research databases; additionally, no discovery campaigns from the biotech/pharma industry have been published to our knowledge.

Any other indications of modulating CB2 receptor being pursued? and the potential risk/benefit of targeting CB2 receptor for obesity? 

  • CB2 receptor modulation has been investigated for the treatment of cancer, addiction disorders, pain management, and metabolic disorders, including obesity. Some benefits of targeting CB2 receptor for the treatment of obesity are avoiding adverse effects from CB1 receptor modulation (e.g., depression and anxiety), and anti-inflammatory effects, given the presence of CB2R in immune cells.

Reviewer 2 Report

Comments and Suggestions for Authors

1.       In table 1 the authors show risk of bias assessment and refer this to the SYRCLE tool. However, one should be able to comprehend this review without consulting the respective paper (SYRCLE). So, the meaning of Q1 – Q10 should be explained here.

2.       In 3.3.2. the authors cite the study by Alshaarwy and conclude that CB2 or CB1/CB2 double ko protects mice from DIO. However, CB2 ko mice were not protected so in this setting (8wk old mice) CB1 seems to be solely responsible for the phenotype but not CB2. This was also the conclusion of the authors.

3.       The study by Schmitz should not be included in this overview as the title here states “regulation of food intake”. Schmitz et. al., however, show that the effect of CB2 is mediated by the interplay of the immune system with adipose tissue and does not include a central component of food intake. This was underlined by their findings that weight gain under HFD is not influenced by Hu308.

4.       The cited study by Dias-Rocha shows hypothalamic CB2. To my knowledge, CB2 is not expressed in the Hypothalamus (except microglial cells), so the results from Dias-Rocha should be interpreted carefully. Also, CB2 antibodies are notorious for unspecific binding and the study does not show whole blots.

5.       In the discussion, the authors state that there is compelling evidence that CB2 influences food intake. In fact, not really. It shows an effect of CB2 on body weight (gain). The results from studies cited show results promoting or negating the hypothesis. Also, as mentioned above the cited studies should be interpreted with much care as other confounders than those in SYRCLE come into play: antibody specificity, use of ligands etc. AM630, SR144528, JWH015 (significant agonist effect on CB1!) all have significant off-target effects making data interpretation difficult.

6.       In CB2 ko mice, there could be compensatory upregulation of CB1 or other receptors/pathways, which is usually not tested.

7.       Also, HFD induced increases in CB2 could be an indirect effect of an altered endocannabinoid tone or a counterregulatory loop initiated by dopamine to control its own release.

Comments on the Quality of English Language

is ok. some minor errors.

Author Response

Dear Reviewer,

Thank you for your valuable feedback on the paper. Regarding your observations, the following changes and clarifications have been included in the manuscript:

In table 1 the authors show risk of bias assessment and refer this to the SYRCLE tool. However, one should be able to comprehend this review without consulting the respective paper (SYRCLE). So, the meaning of Q1 – Q10 should be explained here.

  • The 10 questions included in the SYRCLE bias tool have been included in the manuscript.

In 3.3.2. the authors cite the study by Alshaarwy and conclude that CB2 or CB1/CB2 double ko protects mice from DIO. However, CB2 ko mice were not protected so in this setting (8wk old mice) CB1 seems to be solely responsible for the phenotype but not CB2. This was also the conclusion of the authors.

  • Changes have been made in 3.3.2 and to the conclusion on the article regarding this study to improve clarity on the interpretation of the results.

The study by Schmitz should not be included in this overview as the title here states “regulation of food intake”. Schmitz et. al., however, show that the effect of CB2 is mediated by the interplay of the immune system with adipose tissue and does not include a central component of food intake. This was underlined by their findings that weight gain under HFD is not influenced by Hu308. 

  • In the discussion, results that were mentioned from HU308 administration from Schmitz’s study were eliminated given that they did not influence food intake. However, the study does evaluate differences between wild-type and CB2 KO mice on body weight and food intake, results that are according to the aim of the present systematic review. Therefore, these results are included in the review. Additionally, even though the study focuses on the interplay of the immune system with adipose tissue and does not identify significant differences in endogenous ligands (EA, OEA, PEA, and 2-AG) in the brain, the results provide novel information on the CB2 peripherical mediation regulated by the immune system. Additionally, Schmitz does document changes at the peripheral level in ghrelin and immune cells (IL-6), which do play a role in food intake, suggesting that CB2 is involved in regulating food intake via an interplay between central and peripheral signaling.

The cited study by Dias-Rocha shows hypothalamic CB2. To my knowledge, CB2 is not expressed in the Hypothalamus (except microglial cells), so the results from Dias-Rocha should be interpreted carefully. Also, CB2 antibodies are notorious for unspecific binding and the study does not show whole blots.

  • Details on the results of Dias-Rocha’s study and the authors’ conclusion have been added to the results section in order to provide more clarity for the interpretation of the results.

In the discussion, the authors state that there is compelling evidence that CB2 influences food intake. In fact, not really. It shows an effect of CB2 on body weight (gain). The results from studies cited show results promoting or negating the hypothesis. Also, as mentioned above the cited studies should be interpreted with much care as other confounders than those in SYRCLE come into play: antibody specificity, use of ligands etc. AM630, SR144528, JWH015 (significant agonist effect on CB1!) all have significant off-target effects making data interpretation difficult.

  • This observation has been added as a potential limitation of the present systematic review in the last paragraph of the discussion.

In CB2 ko mice, there could be compensatory upregulation of CB1 or other receptors/pathways, which is usually not tested.

  • This observation has been added as a potential limitation of the present systematic review in the last paragraph of the discussion.

Also, HFD induced increases in CB2 could be an indirect effect of an altered endocannabinoid tone or a counterregulatory loop initiated by dopamine to control its own release.

  • This observation has been added as a potential limitation of the present systematic review in the last paragraph of the discussion.

Round 2

Reviewer 1 Report

Comments and Suggestions for Authors

Thanks for addressing all the comments. The manuscript is well-suited for publication in IJMS.  

Comments on the Quality of English Language

Good.